# Transferrin-Enabled Blood–Brain Barrier Crossing Manganese-Based Nanozyme for Rebalancing the Reactive Oxygen Species Level in Ischemic Stroke

**DOI:** 10.3390/pharmaceutics14061122

**Published:** 2022-05-25

**Authors:** Qianqian Zhao, Wenxian Du, Lingling Zhou, Jianrong Wu, Xiaoxing Zhang, Xiaoer Wei, Sijia Wang, Yu Huang, Yuehua Li

**Affiliations:** 1Department of Radiology, Shanghai Jiao Tong University Affiliated Sixth People’s Hospital, Shanghai Jiao Tong University School of Medicine, 600 Yi Shan Road, Shanghai 200233, China; zhaoqianqian199010@gmail.com (Q.Z.); wx0910@mail.ustc.edu.cn (W.D.); llzhou21@m.fudan.edu.cn (L.Z.); dr.zhang8967@gmail.com (X.Z.); weixiaoer13774356395@gmail.com (X.W.); sjwang121@gmail.com (S.W.); yuhuang6y@163.com (Y.H.); 2Department of Ultrasound in Medicine, Shanghai Jiao Tong University Affiliated Sixth People’s Hospital, Shanghai Jiao Tong University School of Medicine, 600 Yi Shan Road, Shanghai 200233, China; wujr_028@126.com

**Keywords:** ischemic stroke, reactive oxygen species, nanozyme, blood–brain barrier, magnetic resonance imaging

## Abstract

(1) Background: Acute ischemic stroke (IS) is one of the main causes of human disability and death. Therefore, multifunctional nanosystems that effectively cross the blood–brain barrier (BBB) and efficiently eliminate reactive oxygen species (ROS) are urgently needed for comprehensive neuroprotective effects. (2) Methods: We designed a targeted transferrin (Tf)-based manganese dioxide nanozyme (MnO_2_@Tf, MT) using a mild biomimetic mineralization method for rebalancing ROS levels. Furthermore, MT can be efficiently loaded with edaravone (Eda), a clinical neuroprotective agent, to obtain the Eda-MnO_2_@Tf (EMT) nanozyme. (3) Results: The EMT nanozyme not only accumulates in a lesion area and crosses the BBB but also possesses satisfactory biocompatibility and biosafety based on the functional inheritance of Tf. Meanwhile, EMT has intrinsic hydroxyl radical-scavenging ability and superoxide-dismutase-like and catalase-like nanozyme abilities, allowing it to ameliorate ROS-mediated damage and decrease inflammatory factor levels in vivo. Moreover, the released Mn^2+^ ions in the weak acid environment of the lesion area can be used for magnetic resonance imaging (MRI) to monitor the treatment process. (4) Conclusions: Our study not only paves a way to engineer alternative targeted ROS scavengers for intensive reperfusion-induced injury in ischemic stroke but also provides new insights into the construction of bioinspired Mn-based nanozymes.

## 1. Introduction

Stroke is a neurological disorder caused by cerebral ischemia or hemorrhage, and it ranks second in the mortality rate of all diseases [1,2]. Ischemic stroke (IS) accounts for ~87% of stroke cases and is caused by cerebral ischemia and hypoxia due to insufficient blood supply upon embolism or thrombosis [3,4]. In response to ischemia, microglia are activated to protect neuron cells. However, the overactivation of microglia generates inflammatory cell factors, such as interleukin-1β (IL-1β), interleukin-6 (IL-6), and tumor necrosis factor-α (TNF-α), which results in deleterious inflammation and neuronal death [5,6]. After ischemia–reperfusion, significant amounts of reactive oxygen species (ROS), including superoxide anions (O_2_^•−^), hydrogen peroxide (H_2_O_2_), and hydroxyl radicals (•OH), are produced [7,8]. ROS cause cell death via lipid peroxidation at the cellular level, along with oxidative DNA damage, protein damage, and cytoskeletal structural injury [9,10]. Free radical scavengers, which are significant neuroprotective agents, can safeguard neurons against ROS produced in chain reactions after IS [11,12]. For example, edaravone (Eda), the first novel free radical scavenger approved in Japan, was used to treat IS [13]. However, it is subject to several critical problems, such as a short half-life, poor blood–brain barrier (BBB) penetration, and mediocre specificity [14,15]. Accordingly, nanoparticle-mediated ROS-scavenging strategies have been proven for the treatment of diseases induced by oxidative damage [16,17]. Among these nanoparticles (NPs), artificial NPs with intrinsic enzyme-like activities, termed nanozymes, have attracted considerable attention [18,19]. Therefore, there is an urgent need to develop a multifunctional nanozyme that can effectively cross the BBB and scavenge ROS to treat IS so as to relieve oxidative stress and restore neuronal activity.

The BBB is an exclusive biological barrier and the greatest impediment to the protection of the brain from potentially hazardous substances in the blood, hindering the passage of nanozymes and their sufficient accumulation in the brain when IS occurs [20,21]. Numerous brain-targeted NP-delivery strategies have been investigated as ways to increase the efficiency of BBB penetration, among which receptor-mediated transcytosis (RMT) has been extensively researched [22,23,24]. The BBB consists of microvascular endothelial cells, providing substantial potential for exploration with less tissue damage than that associated with other more invasive strategies [25,26]. Transferrin receptor 1 (TfR), the ligand of which is transferrin (Tf), is overexpressed in the BBB and thus very promising for clinical transformation employing RMT mechanisms [26,27,28]. Holo-transferrin, a member of the Tf family that contains two Fe atoms and exhibits a higher affinity for TfR1 than apo-Tf (Tf without bound Fe) and monoiron-Tf (containing one Fe atom), could be extensively used as a distinct ligand for brain-targeted imaging and treatment [29]. Several studies have demonstrated that Tf can combine with organic as well as inorganic NPs [30,31,32]. However, several problems remain to be addressed. The ultrasmall size of NPs provides them with the ability to cross the BBB, but their short vascular circulation time is detrimental to the treatment of stroke [33]. Furthermore, their complex synthesis and preservation of their structural integrity during fabrication are problematic issues [34,35,36]. Therefore, to fully exploit the features of Tf, strategies to maintain its inherent capability to penetrate the BBB and to simplify the synthetic processes required to fabricate Tf-based nanoplatforms are urgently needed.

Manganese (Mn) is essential for various life processes, including the growth of nerves and the development of cognitive function. In recent years, the use of manganese dioxide (MnO_2_) in nanoscience [37,38], biosensors [39,40], bioimaging [41,42,43], drug delivery [44,45,46], and cancer therapy [42,47,48,49,50] has emerged as a highly active research field. MnO_2_ has been demonstrated to exhibit catalase (CAT)- and superoxide dismutase (SOD)-mimicking activities, making it a promising free radical scavenger in the treatment of various brain diseases as a means to prevent neurons from being damaged [51,52]. Furthermore, in specific acidic and H_2_O_2_-rich microenvironments, MnO_2_ is decomposed into Mn^2+^, which can be used for magnetic resonance imaging (MRI) because of a high T_2_ signal [42,53]. Additionally, Mn^2+^, the decomposition product of MnO_2_, shows strong paramagnetic relaxation enhancement ability and thus tremendous potential as an MRI contrast agent [54,55,56]. Compared with normal cells and tissues, more acidic tissues exhibit corresponding time-dependent positive T_1_-MRI signal enhancement, implying that pH changes could be applied to ultrasensitive pH-responsive MRI [57]. Clinical gadolinium (Gd) has its limitations, such as nephrogenic systemic fibrosis, and another imaging research focus, ferrum (Fe), requires further in-depth study. Thus, Mn^2+^, as a promising candidate, was explored due to its excellent properties. The location of acute cerebral infarction is mildly acidic owing to severe glycolysis, with H_2_O_2_ being generated and accumulated. The studies cited above indicate that MnO_2_ could be used for hypersensitive pH-responsive MRI imaging in acute IS treatment. However, the application of facile-fabricated MnO_2_-based multifunctional nanoplatforms in IS treatment remains relatively under-researched.

In this study, we constructed a new nanoplatform based on a MnO_2_ nanozyme, which has strong antioxidant activity and the capacity to effectively remove ROS. As shown in Figure 1, the Tf polypeptide quickly crosses the BBB through endocytosis to reach the lesion area, effectively targeting the site of cerebral infarction, eliminating free radicals in a timely manner, and alleviating inflammatory damage to the BBB caused by oxidative stress. Finally, Eda was loaded in the modified oxide by using the absorption of organic macromolecules and intermolecular force to obtain the Eda-MnO_2_@Tf (EMT) nanozyme so as to synergically remove ROS in the stroke region and reduce the level of inflammatory factors, thereby achieving the purpose of neuron protection. Thus, this work demonstrates the application of Mn-based nanozymes in stroke therapy, providing a precedent for the use of nanozymes in the treatment of clinical problems.

## 2. Materials and Methods

### 2.1. Materials

Holo-Tf (≥98%) was obtained from Shanghai Yuanye Bio-Technology Co., Ltd., Shanghai, China. Bovine serum albumin (BSA ≥ 98%) was from Sigma-Aldrich, St. Louis, MO, USA. Manganese chloride tetrahydrate (MnCl_2_·4H_2_O) and sodium hydroxide (NaOH) were purchased from Shanghai Hushi Chemical Co., Ltd., Shanghai, China, and Shanghai Aladdin Biochemical Technology Co., Ltd., Shanghai, China, respectively. Edaravone was acquired from Med Chem Express. Hydrogen peroxide (H_2_O_2_, 30%), methyl violet (MV), and ferrous sulfate heptahydrate (FeSO_4_·7H_2_O, 99%) were obtained from Sinopharm Chemical Reagent Co., Ltd., Shanghai, China.

### 2.2. Synthesis of MT Nanozyme

The synthesis process of MT refers to the previous research [27]. Holo-Tf (120 mg, 1 mg mL^−1^) was dissolved in deionized (DI) water, and MnCl_2_·4H_2_O solution (0.1 M, 1 mL) was then added, followed by vigorous stirring (800 r min^−1^) at room temperature for ~20 min. Then, NaOH solution (0.05 M, 3.6 mL) was added dropwise to maintain a solution pH = 8.4 and monitored by a pH meter. The reaction proceeded at room temperature for another 30 min to obtain MT nanozyme, followed by dialysis against DI water using a 3.5 kDa cutoff bag, changing the water every 6 h. After 2 d, the solution was concentrated for further use.

### 2.3. Preparation of EMT Nanozyme

Eda was added to the MT nanozyme solution obtained above and stirred overnight. Then, the resultant solution was subjected to dialysis for one day in the same way as mentioned above.

### 2.4. Preparation of MnO_2_@BSA (MB)

The procedure was almost the same as that for MT, except that Holo-Tf was substituted with BSA.

### 2.5. Characterization

Transmission electron microscopy (TEM, JEOL, JEM-2100F electron microscope, Tokyo, Japan) images with different multiples of field of view were obtained to analyze the morphology and structure of the EMT nanozyme. At the same time, an electronic picture of the EMT nanozyme was taken by mobile phone. High-resolution transmission electron microscopy (HRTEM) pictures were also obtained to see the structure distinctly. In addition, energy-dispersive X-ray spectroscopy (EDS) element line scanning was performed to make sure that element Mn was effectively loaded. X-ray photoelectron spectroscopy (XPS, Thermo ESCALAB 250Xi, Thermo Fisher Scientific, Waltham, MA, USA) was used to analyze the elemental valence of Mn. Further, the hydrodynamic size and zeta-potential were tested by dynamic light scattering (DLS, Malvern Zetasizer Nanoseries ZS90, Marvin, UK). Importantly, to confirm that the structure of Tf was not destroyed in the process of synthesizing the EMT nanozyme, Chirascan circular dichroism spectroscopy (CD, Applied PhotoPhysics Ltd., Leatherhead, UK) was applied. After that, Fourier transform infrared spectroscopy (FTIR, Alpha Ⅱ, Bruker, Germany) was further used to demonstrate the loaded Eda. Meanwhile, UV-vis spectrophotometry was performed to make sure that Eda was loaded successfully on MT too. Thermogravimetric analysis (TA, Q500) was used to test the amount of Eda that was lost under high-temperature calcination to determine the loading number of Eda. Finally, the release of Eda over time at different pH conditions was also assessed.

### 2.6. Evaluation of O_2_ Production In Vitro

The quantitative detection of O_2_ was performed using a jpbj-609l portable dissolved oxygen tester. Briefly, Eda, MT (50 ppm), and EMT (50 ppm) were added to H_2_O_2_ (400 µM) solution with a pH of 6.4, and O_2_ production was determined.

### 2.7. Evaluation of Hydroxyl Free Radicals through UV-Vis Spectrophotometry by the Methyl Violet (MV) System 

The hydroxyl radical-scavenging properties of the EMT nanozyme were further monitored by UV-vis spectrophotometry with an MV and Fenton reagent system. The samples contained 0.012 mM MV, 1.0 M H_2_O_2_, and a suitable amount of nanozyme, which were suspended in 5 mL of phosphate buffer solution (PBS, pH = 7.4). The samples were incubated in the dark for several minutes, and then the UV absorption of the solution containing NPs was detected. The oxidation resistance of the system was appraised by comparing the maximum absorbance of MV in the original solution with that after the addition of the EMT nanozyme.

### 2.8. ESR Detection of Hydroxyl Radicals

Hydroxyl radicals exist in the microenvironment of stroke and aggravate AIS. DMPO is the capture agent of hydroxyl radicals, which can capture them immediately. DMPO was dissolved in DI water with a final concentration of 100 mM. Groups were set as follows: control group (DMPO), experimental group (Eda, MT, and EMT + DMPO) with concentrations of 50 ppm, and EMT group (10, 20, 40, and 80 ppm + DMPO). The ESR spectra were collected using a jeol-fa200 paramagnetic spectrometer.

### 2.9. Evaluation of •OOH through ESR Spectroscopy 

The SOD-like enzymatic activity of the EMT nanozyme was investigated by ESR spectroscopy. First, 71 µg of KO_2_ and 1 mg of BMPO were added to methanol (200 µL, 5 mM). Then, different groups, including control, Eda, MT (50 ppm), and EMT (50 ppm), were added to the above system to evaluate the performance of the SOD-like enzyme.

### 2.10. Cytotoxicity Test In Vitro 

Human umbilical vein endothelial cells (denoted as HUVECs, Shanghai Institute of Cells, Chinese Academy of Sciences) were cultured in high-glucose DMEM (Hyclone, Logan, UT, USA) with 10% fetal bovine serum and 1% penicillin at 37 ℃ under 5% CO_2_. Then, the HUVECs were digested with 0.25% EDTA (Gibco, ThermoFisher, Shanghai, China) containing trypsin for 1 min. After counting, the cells were set in a 96-hole plate at 105 cells/well. EMT at different concentrations (0, 12.5, 25, 50, and 100 ppm) was added to the wells. After co-incubation for 12 or 24 h, the original medium was exchanged with fresh medium. Cell viabilities were determined by standard CCK-8 assays (Shanghai Ruicheng Bio-Tech Co., Ltd., Shanghai, China). Absorbance was measured using a standard microplate reader (BioTeck Instrument, Winooski, VT, USA) at a wavelength of 450 nm after 80 min.

### 2.11. Evaluation of the Therapeutic Effect of Cell Level with Calcein-AM/PI

First, cells were plated on a special culture dish at a cell density of 5 × 10^3^ per well for 12 h. Then, H_2_O_2_ (400 µM) was added to the wells. After 2 h, the reagents for the different groups (Eda, MT, and EMT) were dispersed in DMEM and co-incubated with the cells for 4 h. In order to dye the living and dead cells, 100 µL of Calcein-AM and 100 µL of PI solutions were added to the wells. After 15 min, the results of cell death staining by Calcein-AM (green) and PI (red) were observed by a confocal laser scanning microscope (CLSM).

### 2.12. Biodistribution of EMT Nanozyme In Vivo 

Healthy male SD rats weighing 220–250 g (purchased from Shanghai SLAC Laboratory Animal Co., Ltd., Shanghai, China) were used in this experiment. The operation of animal experiments was in accordance with the requirements of animal ethics of Shanghai Jiaotong University. Before the experiment, the animals were ventilated in laminar flow at room temperature of 21 ± 2 ℃. The experimental environment was maintained at 60% humidity for one week. EMT NPs were scattered in saline solution after sterilizing with UV light and then were injected into rats by tail vein with a dose of 10 mg/kg. These healthy SD rats were randomly divided into two groups, one of which served as blank control. The other three groups were injected with the same doses of EMT-containing saline solution (10 mg/kg) for 1, 7, and 30 days. After 30 days of normal feeding, blood samples of rats were taken for whole blood analysis and serum biochemical testing, and the main tissues (heart, kidney, liver, spleen, and lung) of the rats were dissected for hematoxylin and eosin (H&E) staining for histopathological analysis. 

### 2.13. MR Imaging In Vivo 

Rats with middle cerebral artery occlusion (MCAO) were divided into three groups: MB, TfR-blocked, and EMT. Nanozyme dispersion was injected via tail vein at 1 h after thrombectomy. The MCAO rats were injected with MB and EMT dissolved in saline (Mn concentration: 0.136 mmol kg^−1^). In the TfR-blocked group, Holo-Tf solution (dissolved in saline, 10 mg kg^−1^, 1 mL) was i.v. injected 12 h before the injection of EMT to saturate the TfR. Then, the brain regions of the processed MCAO rats were examined with a clinical 3.0T MRI (Prisma, Magnetom, Berlin, Germany) and small animal coil (FOV of 60 × 60 mm; slice thickness of 1 mm without slice spacing). To confirm the onset of AIS, the MRI scan sequence included diffusion-weighted imaging (DWI), apparent dispersion coefficient (ADC) map, T_1_-weighted imaging (T_1_WI), and T_2_-weighted intensity (T_2_WI). The diffusion gradient of DWI was set at a b-value of 1000 s mm^−2^. After injecting nanozyme, images were collected at 5 min, 30min, 1 h, 2 h, 3 h, and 6 h after injection. The parameters of T_1_WI are as follows: repetition time (TR)/echo time (TE) = 360/13.2 ms; acquisition matrix = 256 × 160; field of view = 60 × 60 mm; number of slices = 30; slice thickness = 1 mm; flip angle = 1500. 

### 2.14. Statistical Analysis

Data are expressed as the mean or mean ± standard deviation (ns: *p* > 0.05, * *p* < 0.05, ** *p* < 0.01, *** *p* < 0.001). These results were obtained by SPSS software (SPSS, Chicago, IL, USA).

## 3. Results and Discussion

### 3.1. Synthesis and Properties of Eda-MnO_2_@Tf

A MnO_2_ nanozyme was developed in situ on Tf via reformative biomineralization (Figure 1a). Its spherical morphology and bright orange solution are shown in Figure 1b–d. Figure 1b confirms that the MT nanozyme was successfully synthesized. Moreover, Appendix A demonstrates the effective doping of Mn. Inspired by ceria nanoparticles, we hypothesize that the EMT nanozyme can also scavenge free radicals by shifting between the Mn (IV) and Mn (III) forms on the nanoparticle surface. As shown in Figure 1e, the XPS results of EMT reveal the valence states of Mn. The relative contents of Mn^3+^ and Mn^4+^ in the EMT nanozyme were 44.4% and 55.6%, respectively. Meanwhile, the existence of oxygen defects (O_2_) and lattice oxygen (O_3_) were considered to be favorable for the catalytic reaction (Figure 1f). Moreover, the relative contents of adsorbed oxygen, oxygen defects, and lattice oxygen were 21.7%, 73.5%, and 4.8%, respectively (Appendix A). The EMT nanozyme has good dispersion and uniform size, which helps it to cross the BBB. As shown in Figure 1g, the average size of the EMT nanozyme is about 50 nm, and its polydispersity index is 0.231. This figure also demonstrates that loading with Eda had almost no effect on the particle size of the nanozyme. However, its ζ-potential was lower after Eda loading (Appendix A). The biomineralization strategy in this study was the addition of Tf, the structure of which is easily changed, leading to inactivation. Accordingly, the secondary structure of Tf was investigated using CD spectroscopy. Figure 1h shows that the CD spectrum of EMT is the same as that of Tf, indicating that the performance of Tf and its ability to cross the BBB were maintained during EMT synthesis. In addition, the FTIR spectra clearly show that the stretching vibration peaks for the EMT nanozyme at 1466, 1390, 1247, 1017, and 1030 cm^−1^ correspond to those for Eda (Appendix A). The UV-vis analysis results also confirm the efficient loading of Eda (Appendix A). TGA was conducted to determine the amount of Eda loaded onto the EMT nanozyme, and the results show that 28.8% of the weight of MT was preserved upon a temperature increase to 800 °C, while that for EMT was 19.0%. It is worth noting that Eda was added to the original reaction solution, so the Eda loading efficiency was calculated to be 9.8% according to the TGA results (Figure 1i). To a certain extent, the focal area of stroke is hypoxic, and its microenvironment is acidic. Accordingly, the pH-responsive release of Eda was investigated (Figure 1j). The results show that the release of Eda increased with the decrease in pH, and there was almost no release in the neutral environment (pH = 7.4). This indicates that the in vivo level of Eda can be effectively increased in the lesion area without significant impact on normal physiological tissues.

As shown in Figure 2a, the enzyme-like properties of the EMT nanozyme were investigated, which revealed the ability to remove ROS. MV was used as a typical reagent for detecting •OH. The characteristic absorption peak of MV (Figure 2b) shows that EMT has a stronger ability to scavenge •OH than Eda and MT, confirming its ability to scavenge •OH at the cellular and in vivo levels. Furthermore, Figure 2c evaluates the oxidation resistance of EMT at different concentrations of 20 ppm, 40 ppm, and 80 ppm. The ESR results confirm this conclusion (Figure 2d,e). The characteristic •OH peak for the EMT nanozyme has the lowest intensity for the 1:2:2:1 pattern, and the higher the concentration of the EMT nanozyme, the stronger the ability to remove •OH. As shown in Figure 2f, the EMT nanozyme has SOD-like activity and can convert superoxide anions (O_2_^•−^) with strong oxidative activity into O_2_. Moreover, we further explored the SOD-like properties of EMT NPs by analyzing the ESR spectrum. O_2_^•−^ is usually unstable in water and exists in the form of •OOH. As shown in Figure 2g, characteristic peaks at 1:1:1:1 in the ESR spectrum indicate the presence of •OOH in the control group. Compared with the control group, corresponding •OOH concentrations were reduced significantly in the EMT groups, which indicates that the EMT nanozyme possesses excellent SOD-like performance (Figure 2h). Meanwhile, the CAT-like activity of the EMT nanozyme was also studied. Both MT and EMT produced oxygen, but EMT did so more efficiently. However, Tf and Eda produced almost no oxygen. Thus, it can be inferred that the CAT-like enzyme activity of EMT is mainly attributed to the excellent CAT-like catalytic activity of MnO_2_ (Figure 2i). 

### 3.2. MRI Performance of EMT In Vitro

MRI is typically applied in the diagnosis of IS owing to its outstanding spatial resolution, and the use of contrast agents enables the observation and assessment of the stroke area more clearly. The current clinical contrast agent Gd-DTPA suffers from comparatively low susceptibility and latent side effects. Therefore, to evaluate the response release of Mn^2+^ and the potential use of the EMT nanozyme as a contrast agent, the MRI performance of EMT in vitro was explored using a simulated stroke microenvironment. T_1_ images of various concentrations of the EMT nanozyme were obtained. As shown in Figure 3a, as the concentration of Mn increased from 0 to 0.2 mM, the MR signal also increased, suggesting that the nanozyme generates a high magnetic field gradient. The longitudinal relaxation rate (r1) for the EMT nanozyme was different under different conditions. With decreasing pH, r1 increased from 2.81 to 6.98 mM^−1^s^−1^, evidencing a clear pH-responsive enhancement for MR imaging. Compared with the results for pH 7.4, the r1 value of which was only 2.81 mM^−1^s^−1^, the r1 value increased to 6.98 and 5.86 mM^−1^s^−1^ for the solutions with pH values of 5.0 and 6.4, respectively. Furthermore, Figure 3d illustrates that, as the concentration of H_2_O_2_ increased from 100 to 200 and 400 µM at pH = 6.4, r1 increased correspondingly, suggesting that the presence of H_2_O_2_ dramatically promotes the transformation of Mn^4+^ to Mn^2+^.

Under slightly acidic and higher H_2_O_2_ concentration conditions, the corresponding in vitro MR images appear brighter with increasing Mn concentration (Figure 3b,c,e,f). 

Similarly, at pH = 7.4, r1 was much lower, which means that water was isolated from Mn, negating its effect on longitudinal relaxation time [58]. Moreover, compared with small molecules, the EMT nanozyme has a comparatively large molecular mass, which makes a greater contribution to shortening the rotational tumbling time and helps to obtain a fairly good r1 value [59].

### 3.3. BBB Penetration In Vitro and Cellular Uptake

Before performing cellular uptake and BBB-crossing assays, we used counting kit-8 (CCK-8) assays to assess the cytotoxicity of the material. As shown in Figure 4a, HUVEC cells presented over 90% viability after incubation with up to 100 µg mL^−1^ EMT nanozyme for 24 h, confirming its negligible cytotoxicity. 

The ability to cross the BBB is essential for the EMT nanozyme to play a therapeutic role. Almost all macromolecular drugs and 98% of small molecule drugs have difficulty passing through the BBB to brain lesions. However, Tf, an essential nutrient for the brain, can target and cross the BBB to some extent and enter the brain parenchyma. Because the expression of TfR on the endothelial cell is relatively high, it is an important target receptor for targeting the BBB. Meanwhile, EMT nanozymes exclusively bind to TfR, crossing the BBB effectively through a receptor-mediated transcellular transport mechanism. Accordingly, we established an in vitro BBB model using brain endothelial (bEnd.3) cells to assess the BBB permeability of the nanozyme. As shown in Figure 4b, bEnd.3 cells were cultured on the upper surface of culture inserts in transwell assays, with the transendothelial electrical resistance (TEER) measured every day. Tight contact among the developing cells, confirmed by the TEER value reaching 200 Ω cm^2^, indicated that the BBB model was successfully established. The integrity and compactness of the bEnd.3 cells are sufficiently comparable to the in vivo BBB [60]. The amounts of FITC-labeled EMT nanozyme transferred from the upper chamber to the lower chamber were detected to assess BBB permeability at different times. As shown in Figure 4c, the BBB permeation of EMT reached 15.1% after incubation for 4 h, further demonstrating that Tf was well preserved during EMT synthesis and retained the targeting properties of Tf-based materials. In order to investigate the mechanism by which the Tf ligands on the EMT nanozyme specifically bind to overexpressed TfR1 receptors in the BBB, thus mediating its crossing, we used a typical blocking experiment. A large dose of free Tf was put into the upper chamber medium 12 h in advance to saturate the TfRs, blocking the binding of EMT to TfRs on the cell surfaces before they were added at a concentration of 100 µg mL^−1^. The results show that the BBB-crossing efficiency for the Blocked + EMT group was significantly decreased to 9.5%, which was dramatically lower than that for the EMT group.

The overexpressed TfRs on the surfaces of HUVEC membranes promoted the uptake of EMT, which forms stable complexes with TfRs, unlike the Tf-blocked experimental group. Furthermore, the FITC-labeled EMT nanozyme was used for flow cytometric analysis. The results in Figure 4d show that, with increasing incubation time from 1 to 8 h, the green fluorescence intensified in both groups, indicating increased absorption of the nanozyme. After 4 h of incubation with HUVEC cells, the fluorescence signal ratio was 1.85 times higher for the EMT group than that for the Tf-blocked group, confirming that blocking the TfRs greatly hinders the absorption of EMT, indicating the importance of TfRs as its transport medium. Through the above experiments, we can conclude that the targeting material can quickly cross the BBB and shows enhanced brain uptake in a receptor-mediated manner, ensuring its neuroprotective effect in the lesion area. 

### 3.4. HUVEC Protection and ROS Scavenging In Vitro

To evaluate the neuroprotective effect of EMT against oxidative damage in ischemic brain tissues, which results from the overproduction of ROS, we used a common in vitro model based on HUVEC cells, exposing them to H_2_O_2_ (400 µM) for 2 h. As shown in Appendix A, the mortality rate was inapparent after incubating for 0.5 h, while the cell viability decreased significantly with the extension of incubation time to 1 h, 2 h, and 4 h. After adding the EMT nanozyme, compared with the 1 h and 4 h groups, the 2 h group showed the best therapeutic effect. Moreover, to differentiate the treatment outcomes of Eda, the MT nanozyme, and the EMT nanozyme, cell viabilities were inevitably tested. As shown in Figure 4e, compared with 17.7% for the H_2_O_2_ control group, the cell survival rates for the Eda and MT groups were noticeably higher at 32.13% and 61.68%, respectively, while the viability for the EMT group was 67.9%, which indicates the significant neuroprotective effect of the EMT nanozyme under oxidative conditions. This confirms the importance of Tf targeting in improving cellular uptake, which elevates ROS-scavenging activity. Furthermore, we tested the capability of the EMT nanozyme to ameliorate oxidative-stress-mediated injury, with its concentration varying from 25 to 100 µg mL^−1^ (Figure 4f). The overall survival was only 12% for the H_2_O_2_ group. However, the survival rate increased from 22.3% to 66.9% with increasing EMT concentration. 

This efficient intracellular ROS removal by the EMT nanozyme was confirmed by the numbers of living and dead cells stained with Calcein-AM (green) and PI (red), respectively (Figure 4g). In the Calcein-AM/PI assays, the effects of the EMT nanozyme on HUVEC cells can be directly observed by CLSM detection of living and dead cells. We found that there were a large number of dead cells in the control groups, while the number of dead cells was significantly decreased in the presence of EMT and H_2_O_2_, as evidenced by weak red and strong green signals, indicating that the introduction of TfRs promoted the endocytosis of EMT. In this process, the Mn-encapsulating Tf competitively binds to the receptors on the surface of the HUVECs, promoting the therapeutic effect of the EMT nanozyme for stroke.

The fluorescent compound 2,7-dichloro-dihydrodiacetate (DCFH-DA) is a kind of ROS probe. Here, non-fluorescent DCFH is oxidized to 2,7-dichlorofluorescein (DCF), which emits green fluorescence, upon reaction with ROS. CLSM was used to evaluate the average intracellular fluorescence density in cells so as to verify the mechanism by which EMT enhances therapeutic efficiency (Appendix A). Significantly higher ROS levels were observed inside cells after treatment with H_2_O_2_. Notably, the green fluorescence for the EMT group was the weakest among all of the groups, which is consistent with the results of ESR characterization in vitro. We used Image Pro Plus to analyze the fluorescence intensity, as shown in Figure 4h, which once again evidenced the ROS-removing capability of EMT. Moreover, flow cytometry results were used to directly show the quantitative level of intracellular ROS to prove the antioxidative stress ability of the EMT nanozyme. As expected, with the increase in the EMT concentration, the intracellular level of total ROS became lower. When the EMT concentration was 25 ppm, 50 ppm, and 100 ppm, the corresponding fluorescence intensity was 26.17%, 12.96, and 8.53%, respectively (Appendix A). The decrease in fluorescence intensity is due to the large amount of ROS being scavenged by the EMT nanozyme with excellent SOD- and CAT-mimicking activities and intrinsic •OH-scavenging ability. Thus, the higher the EMT nanozyme concentration, the stronger the antioxidant capacity of scavenging ROS, which is consistent with the ESR results. In addition, the level of fluorescence was reduced in the EMT-treated cells, lower than that of the H_2_O_2_, Eda, and MT groups, which indicates that tremendous amounts of ROS were eliminated by the EMT nanozyme (Appendix A).

### 3.5. Localization and MRI Performance of EMT In Vivo

EMT labeled with Cy 5.5, a red fluorescent dye, was prepared and injected into Sprague Dawley (SD) rats through the tail vein to investigate the in vivo targeting ability of the EMT nanozyme in MCAO rats. As shown in Figure 5a, no fluorescence signals of Cy5.5 (red) were obviously visualized in the sham group, which is consistent with previous studies. In the ischemic lesions of TfR-blocked stroke-bearing rats (i.e., those injected with Tf solution 12 h before EMT injection to saturate the TfRs on the BBB), a slight increase in red fluorescence was observed due to the aggregation of the EMT nanozyme without targeting after ischemic damage. However, the fluorescence signal of the EMT group increased rapidly and reached a peak at 6 h, which was attributed to the effect of Tf targeting. Moreover, we used fluorescence imaging to explore the distribution of Cy5.5-EMT in multiple organs (brain, heart, liver, spleen, lung, and kidney) excised from rats 1, 3, 6, and 12 h after injection. As shown in Appendix A, the fluorescence accumulated in ischemic lesions of the excised brains of rats and reached its peak at 6 h and faded to weak by 12 h. The fluorescence levels in the other organs underwent varying degrees of attenuation with time after reaching their peaks. The fluorescence intensity of the kidney was relatively high, because the kidney is the main metabolic site. These results demonstrate that EMT can effectively penetrate the BBB and specifically bind to damaged neurons for drug release in SD rats. 

Based on previous studies on imaging conditions, we evaluated the BBB permeability of the EMT nanozyme and whether it can accurately reach the stroke-specific (i.e., mildly acidic and hypoxic) microenvironment with the aid of Tf, thus benefiting diagnosis and treatment of IS. Accordingly, two control groups, BSA-MnO_2_ (removal of targeting ability) and TfR-blocked (saturated targeted receptors), were established to further explore Tf targeting capability. After performing MCAO, head MRI imaging of IS rats clearly revealed lesions in the left hemisphere. In non-contrast MR scanning, a moderately high T_2_ signal, a high signal for diffusion-weighted imaging, and a low apparent diffusion coefficient signal were observed for all of the groups (Appendix A), clearly identifying the specific lesion location and size. For the TfR-blocked group, no obvious enhancement was observed immediately, but the focus was slightly enhanced after 30 min. At 1 h, the lesion was enhanced significantly, and the enhancement gradually weakened, disappearing over time, which means that the loss of Tf targeting removes the BBB-traversing capability. In the BSA-MnO_2_ group, slightly positive contrast enhancement was observed immediately after injection, showing a similar trend to that in the TfR-blocked group, but the intensity in T_1_ imaging was much higher after 6 h. Compared with the two control groups, the infarcted areas in the EMT group presented the brightest enhanced intensity in the T_1_ WI just 1 h after EMT injection, confirming its satisfactory MRI performance. Thus, we can make a clear diagnosis of the infarct in the contralateral cerebral hemisphere by means of EMT, which confirms that the EMT nanozyme possesses the ability to cross the BBB. 

### 3.6. Protection against Stroke and Prevention of BBB Damage In Vivo

With robust evidence of the antioxidative properties of EMT nanozymes in vitro in hand, we explored the in vivo therapeutic effect in ischemic SD brains by tail vein injection. Before injection, rats were subjected to MCAO. Then, 24 h later, the brains were harvested, sliced, and stained with 2,3,5-triphenyltetrazolium chloride (TTC), a common index of cell viability widely used for evaluating cerebral ischemic injury, where red and white staining indicate noninfarcted and infarcted tissue, respectively. In the sham group (Figure 5b), noninfarcted areas were detected, while a massive infarction (41.7%) was detected in the control group (Figure 5c). For rats treated with different doses of the EMT nanozyme, the results (Figure 5d,e) show that the infarct volume decreased from 41.7% to 9.6% as the concentration of Mn increased from 0.4 to 0.8 mg/kg. This indicates that EMT accumulates in the infarct area due to the targeting of Tf, enhancing its therapeutic effect. Rats pretreated with Eda, MT, and EMT were found to be significantly less vulnerable to ischemia at a dose of 0.6 mg/kg. Correspondingly, EMT outperformed the other materials, which is largely due to the ROS-eliminating ability of the Eda loaded in the Tf-targeting system.

An enzyme-linked immunosorbent assay (ELISA) was used for the qualitative and quantitative detection and identification of biomolecules [61]. Studies have shown that Mn can reduce the inflammatory response of tumor lesions. Accordingly, we treated MCAO rats with Eda, MT, or EMT, extracted the protein from the brain tissue, and then evaluated the levels of TNF-α, IL-1β, and IL-6, which are the main inflammatory factors associated with AIS. As shown in Figure 5f, Eda, MT, and EMT all decreased the levels of inflammatory factors to varying degrees, with EMT performing best.

Meanwhile, in order to further investigate the antioxidant mechanism by which EMT protects brain tissue in vivo, blank control and EMT group tissues were stained with DCFH-DA. As discussed above, ROS in brain tissue oxidizes non-fluorescent DCFH to produce red fluorescent DCF. Subsequently, the brain ROS levels were evaluated using confocal laser scanning microscopy (CLMS). Comparing the ROS levels in the brains of the MCAO groups with those of the EMT groups demonstrated that infiltration of the EMT nanozyme expeditiously suppressed the production of ROS (Appendix A). All of the results further demonstrate the potential of the EMT nanozyme in the therapy of IS.

It is essential that EMT protects BBB integrity during IS treatment. Accordingly, to investigate the effect of EMT on BBB damage in ischemic brains, we measured BBB permeability using Evans Blue (EB) staining. As shown in Figure 6a, the right half of the brain presents a bright blue color after ischemic injury, indicating that EB has clearly extravasated across the BBB in MCAO-induced ischemic rats. The measurement of EB exudation can reflect the openness of the BBB, based on which the BBB permeabilities for different groups were explored. When the nervous system is intact (sham), EB combined with plasma albumin cannot pass through the BBB, and no blue color is observed in the brain. Conversely, if the BBB is disrupted, the blue color is observed. Compared with the Eda group and the MT group, the EMT group showed inhibited BBB leakage at a dose of 100 µg/mL. Furthermore, EB perfusion staining combined with CLMS can be used to observe the EB fluorescence intensity in brain slices, which can be used to detect the morphological changes in blood vessels in the BBB and quantitatively analyze the content of EB in brain tissues. As shown in Figure 6b, the weakest red fluorescence was observed in the brain tissue of EMT-treated rats, again demonstrating the effective targeting and accumulation of EMT in the brain and its capacity to eliminate ROS.

### 3.7. In Vivo Biocompatibility

Although the EMT nanozyme has potential biomedical applications and has attracted extensive research interest, its potential toxicity in the treatment of stroke is still unclear. Accordingly, we used healthy SD rats to study the toxicity of the material in vivo. The EMT nanozyme (10 mg/kg) was injected via the tail vein. After several days of normal feeding, the rats were euthanized. Blood samples were collected from the abdominal aorta for routine blood testing and biochemical analysis, and the main organs were dissected for hematoxylin and eosin (H&E) staining analysis. During the observation, it was found that there was no significant change in animal weight (Appendix A). As shown in Appendix A, compared with the healthy rats, the main biochemical indexes and blood indexes of the rats treated with the EMT nanozyme were not significantly different. Further H&E histological analysis (Figure 6c) showed that, compared with the healthy rats, H&E staining results for the main organs (heart, liver, spleen, lung, and kidney) in the experimental group injected with the EMT nanozyme showed no obvious acute or chronic pathological toxicity or adverse reactions. These results preliminarily demonstrate that the EMT nanozyme is biologically safe. 

## 4. Conclusions

In this work, we designed a Tf-based EMT nanozyme to cross the BBB, and we demonstrated its excellent brain targeting in vitro and in vivo. Furthermore, EMT can mimic the activities of CAT and SOD enzymes to effectively eliminate ROS, thereby protecting neuronal cells from the effects of oxidative stress and reducing the levels of inflammatory factors in the lesion area. In addition, EMT responds to the weakly acidic microenvironment in the IS lesion area, which also has a certain concentration of H_2_O_2_, to achieve excellent imaging capabilities. Overall, our study offers a new in situ synthetic method for theranostic nanomaterials. It also demonstrates the viability of integrated ROS removal, treatment, and visualization for reperfusion-induced injury in IS.

## Data Availability

The datasets and materials used in the study are available from the corresponding author.

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
