# Peer review of "Transferrin-Enabled Blood–Brain Barrier Crossing Manganese-Based Nanozyme for Rebalancing the Reactive Oxygen Species Level in Ischemic Stroke"

_pharmaceutics, 2022, doi:10.3390/pharmaceutics14061122_

Round 1

Reviewer 1 Report

In this manuscript the authors report the development of a targeted transferrin-based manganese dioxide nanozyme for rebalancing the reactive oxygen species level and providing neuroprotective effects against ischemic stroke. The nanozyme accumulates in lesion areas and crosses the blood-brain barrier, and the released manganese ions in the weak acid environment of the lesion area can be used for MRI to monitor the treatment process. I suggest that the manuscript could be suitable for publication in Pharmaceutics if the text is subjected to a large number of minor corrections and changes, as follows:

Line 1.  Change “Type of the Paper (Article)” to “Article”

Line 3.  Insert “the” before “ROS” and consider changing “ROS” to “Reactive Oxygen Species”

Line 2.  Remove “Title”

Line 15.  Define “BBB”

Line 19.  Insert “the” after “obtain”

Line 21.  Change “cross” to “crosses” and insert “on” after “based”

Line 24.  Insert “the” before “weak”

Line 25.  Insert “the” before “lesion”

Line 27.  Change “scavenger” to “scavengers”

Line 40.  Change “ROS” to “reactive oxygen species (ROS)”

Line 42.  Change “causes” to “cause”

Line 47.  Insert “a” before “short” and change “BBB” to “blood-brain barrier (BBB)”

Line 72.  Change “simplifying” to “to simplify”

Line 82.  Insert “a” before “high”

Line 84.  Change “magnetic resonance imaging (MRI)” to “MRI”

Line 96.  Insert “a” before “MnO2”

Line 97.  Change “have” to “has”

Line 102.  Change “obtain EMT” to “obtain the Eda-MnO2@Tf (EMT)”

Lines 108-109.  Rewrite “Schematic illustration of EMT nanozyme possessing the function including across the BBB, ROS-relieving capacity, eliminating inflammation and MR Imaging function.”

Line 111.  Remove the full-stop after “Materials”

Line 112.  Change “Holo-Transferrin (Tf, ≥ 98%)” to “Holo-Tf (≥ 98%)”

Line 113.  Change “were bought” to “was from”

Line 119.  Remove the full-stop after “[27]”

Line 127.  Remove the full-stop after “nanozyme”

Line 129.  Change “1 d” to “one day”

Line 131.  Remove the full-stop after “(MB)”

Line 133.  Change “bovine serum albumin (BSA)” to “BSA”

Line 134.  Remove the full-stop after “Characterization”

Line 136.  Change “operated” to “obtained”

Line 137.  Insert “the” before “EMT” (x2)

Line 141.  Change “is” to “was”

Line 142.  Change “studied” to “used” and insert “the” before “hydrodynamic”

Line 145.  Insert “the” before “EMT” and change “spectrometer” to “spectroscopy”

Line 146.  Change “spectra” to “spectroscopy”

Line 147.  Insert “spectrophotometry” after “UV-vis”

Line 149.  Change “investigated” to “used” and “Eda was” to “Eda that was”

Line 152.  Remove the full-stop after “in vitro”

Line 156.  Change “Evaluation of •OH through the UV-Vis spectroscopy by Methyl violet (MV) system” to “Evaluation of hydroxyl free radicals through UV-vis spectrophotometry by the methyl violet (MV) system” and remove the full-stop after “system”

Line 157.  Insert “the” before “EMT and change “detected” to “monitored”

Line 158.  Change “UV-Vis spectroscopy” to “UV-vis spectrophotometry”

Line 160.  Change “Incubated” to “The samples were incubated”

Line 161.  Change “the UV” to “then the UV”

Line 163.  Insert “the” before “EMT”

Line 164.  Remove the full-stop after “Radicals”

Lines 166-167.  Change “hydroxyl radical, which can capture the hydroxyl radical immediately.” to “hydroxyl radicals, which can capture them immediately.”

Line 167.  Change “Dissolve DMPO powder in” to “DMPO powder was dissolved in”

Line 170.  Change “by” to “using a”

Line 171.  Remove the full-stop after “spectroscopy”

Line 172.  Insert “the” before “EMT”

Line 173.  Change “Methanol” to “methanol”

Line 175.  Insert “the” before “above” and insert “the” before “SOD”

Line 176.  Remove the full-stop after “in vitro”

Line 177.  Remove “Cytotoxicity testing in vitro.”

Line 185.  Change “To measure the absorbance, we used a standard” to “Absorbance was measured using a standard”

Line 187.  Insert “the” before “therapeutic” and remove the full-stop after “AM/PI”

Line 188.  Remove “Evaluation of therapeutic effect at the cell level with Calcein-AM/PI.”

Line 194.  Insert “a” before “confocal”

Line 195.  Change “in Vivo” to “in vivo” and remove the full-stop after “vivo”

Lines 197-198.  Change “experiment” to “experiments”

Line 201.  Change “ultraviolet” to “UV light”

Line 207.  Insert “the” before “rats”

Line 209.  Remove the full-stop after “in vivo”

Line 210.  Define “MCAO”

Line 211.  Remove “group”

Line 213.  Insert “the” before “TfR”

Line 217.  Change “AIS, MRI scan sequence including the diffusion” to “AIS, the MRI scan sequence included diffusion”

Line 219.  Remove “were performed”

Line 224.  Remove the full-stop after “analysis”

Line 228.  Remove the full-stop after “Eda-MnO2@Tf”

Line 229.  Change “MnO2 nanozyme were” to “A MnO2 nanozyme was”

Line 230.  Change “their” to “its”

Line 231.  Insert “the” before “MT”

Line 233.  Insert “the” before “EMT”

Line 235.  Insert “the” before “EMT”

Line 236.  Change “are respectively 44.4% and 55.6%” to “were 44.4% and 55.6%, respectively”

Line 237.  Change “are” to “were”

Lines 238-239.  Change “are respectively 21.7%, 73.5%, and 4.8%” to “were 21.7%, 73.5%, and 4.8%, respectively”

Line 239.  Change “Eda-MnO2@Tf (EMT) nanozyme have” to “The EMT nanozyme has”

Line 246.  Insert “the” before “EMT”

Line 247.  Insert “the” before “EMT” (x2)

Line 248.  Insert “the” before “EMT”

Line 252.  Change “circular dichroism (CD)” to “CD”

Lines 254-255.  Change “Fourier-transform infrared spectra (FTIR)” to “FTIR spectra”

Line 255.  Insert “the” before “EMT”

Lines 257-258.  Change “Thermogravimetric analysis (TGA)” to “TGA”

Line 259.  Change “is” to “was”

Line 260.  Change “is 19” to “was 19”

Line 270.  Insert “the” before “EMT”

Line 271.  Insert “spectra” after absorption”

Line 274.  Change “about” to “for”

Line 275.  Insert “spectra” after absorption”

Line 276.  Change “about” to “for”

Line 279.  Insert “the” before “EMT”

Line 280.  Change “revealing” to “revealed” and “Methyl violet (MV)” to “MV”

Line 285.  Change “electron spin resonance (ESR)” to “ESR”

Line 287.  Insert “the” before “EMT”

Line 288.  Change “EMT nanozyme have” to “the EMT nanozyme has”

Line 291.  Change “Characteristic” to “characteristic”

Line 294.  Insert “the” before “EMT”

Line 295.  Insert “the” before “EMT”

Line 299.  Remove the full-stop after “in vitro”

Line 301.  Insert “the” before “EMT”

Line 311.  Insert “the” before “EMT”

Line 313.  Change “these nanozyme generate” to “the nanozyme generates”

Line 314.  Insert “the” before “EMT”

Line 326.  Change “EMT nanozyme owing” to “the EMT nanozyme has”

Line 329.  Remove the full-stop after “uptake”

Line 332.  Insert “the” before “EMT”

Line 336.  Insert “the” before “EMT”

Line 337.  Insert “the” before “in vitro” and change “nanozyme” to “nanozymes”

Line 338.  Insert “the” before “BBB”

Line 339.  Change “demonstrated” to “demonstrating” and “membrane” to “membranes”

Line 341.  Insert “The” before “T test” and insert “the” before “P value”

Line 342.  Insert “The” before “T test” and insert “the” before “P value”

Line 343.  Insert “The” before “T test”

Line 344.  Insert “the” before “P value”

Line 347.  Insert “the” before “EMT”

Line 352.  Insert “the” before “BBB” and change “nanozyme” to “nanozymes”

Line 362.  Change “time” to “times”

Line 370.  Change “is” to “was”

Line 371.  Change “is” to “was”

Line 374.  Change “were” to “was”

Line 377.  Insert “the” before “nanozyme” and change “is” to “was”

Line 383.  Remove the full-stop after “in vitro”

Line 387.  Change “are” to “were”

Line 390.  Change “is” to “was”

Line 393.  Insert “the” before “EMT”

Line 395.  Change “is” to “was”

Line 397.  Change “EMT nanozyme is” to “the EMT nanozyme was”

Line 398.  Change “mounts” to “amounts”

Line 399.  Insert “the” before “EMT”

Line 402.  Remove “death”

Line 405.  Insert “the” before “EMT”

Line 407.  Change “2,7-Dichloro-dihydrodiacetate fluorescent (DCFH-DA)” to “The fluorescent compound 2,7-dichloro-dihydrodiacetate (DCFH-DA)”

Line 411.  Change “are” to “were”

Line 412.  Change “is” to “was”

Line 416.  Change “are” to “were” and insert “a” before “quantitative”

Line 417.  Insert “the” before “EMT”

Line 422.  Insert “that” before “tremendous”

Line 424.  Remove the full-stop after “in vivo”

Line 427.  Insert “the” before “EMT”

Line 431.  Insert “the” before “EMT”

Line 433.  Change “is” to “was”

Line 437.  Change “brain” to “brains”, “reaches” to “reached” and “fades” to “faded”

Line 438.  Change “undergo” to “underwent”

Line 439.  Change “is” to “was”

Lines 425-442.  This paragraph begins by talking about rats, then about mice and then about rats again.  Were all experiments performed on rats or on mice too?

Line 444.  Insert “the” before “EMT”

Line 449.  Change “reveals” to “revealed”

Line 451.  Change “are” to “were”

Line 453.  Change “is” to “was” (x2)

Line 454.  Change “is” to “was” and “weakens” to “weakened”

Lines 456-457.  The figure legend refers to rats and then to mice, which is it?

Line 461.  Insert “the” before “EMT”

Line 462.  Insert “the” before “T test” and insert “the” before “P value”

Line 465.  Change “is” to “was”

Line 467.  Change “is” to “was”

Line 468.  Change “present” to “presented”

Line 471.  Insert “the” before “EMT”

Line 472.  Remove the full-stop after “in vivo”

Line 473.  Change “property” to “properties”, “nanozyme” to “nanozymes” and “in hand” to “to hand”

Line 476.  Change “indexes” to “index”

Line 479.  Change “are” to “were” and “is” to “was” and insert “the” before “control”

Line 480.  Insert “the” before “EMT”

Line 481.  Change “decreases” to “decreased”

Line 482.  Change “increases” to “increased”

Line 485.  Change “outperforms” to “outperformed”

Line 488.  Insert “of” before “biomolecules”

Line 492.  Change “decrease” to “decreased”

Line 499.  Insert “the” before “EMT”

Line 501.  Insert “the” before “EMT”

Line 510.  Insert “the” before “Eda”

Line 511.  Insert “the” before “MT”

Line 515.  Change “is” to “was”

Line 518.  Remove the full-stop after “biocompatibility”

Line 519.  Change “EMT nanozyme have” to “the EMT nanozyme has”

Line 520.  Change “its” to “their”

Line 522.  Change “were” to “was”

Line 528.  Insert “the” before “EMT”

Line 538.  Change “nanozyme show” to “nanozymes showed”

Line 539.  Insert “the” before “EMT”

Line 542.  Change “designed the” to “designed a”

Line 543.  Change “its” to “their”

Reviewer 2 Report

The submitted manuscript "Transferrin-Enabled Blood-Brain Barrier Crossing Manganese-Based Nanozyme for Rebalancing ROS Level in Ischemic 3 Stroke" describes an alternative targeted ROS scavenger for intensive reperfusion-induced injury in ischemic stroke.

Although the outcomes are interesting the manuscript suffers of many flaws that should be corrected before the manuscript becomes acceptable for publication.

1) The authors should clearly explain why they use the term "nanozyme" while no enzyme is included in NPs. This should be stated as soon as in the introduction. Indeed, after reading the manuscript one can understand that "EMT can mimic the activities of CAT and SOD enzymes to effectively eliminate ROS" but in this case the term "nanozyme-like" is more appropriate.

2) All the legends to figures should be revised to give sufficient information to understand the content without referring to the text.

3) All the abbreviations should be defined in the first use (e.g. BBB is not defined).

4) Remove the word "Title" from the title and eventually ajust the term "nanozyme"

Reviewer 3 Report

The article is very intriguing with a concrete future therapy potential and has a clear structure with a well described background too. 

Major issue:

- The authors observed Significantly higher ROS levels inside cells  after treatment with H2O2. Did the authors consider to evaluate other oxidation damages as lipid peroxidation or protein oxidation and their envolvement?

- As pointed out by the authors “the higher the EMT concentration, the stronger the antioxidant capacity, which was attributed to the excellent SOD, CAT mimicking activities and intrinsic •OH-scavenging ability” Could the author can be more accurate on describing this so crucial activities?

-As described “ To evaluate the neuroprotective effect of EMT against oxidative damage in ischemic brain tissues, which results from the overproduction of ROS, we used a common in vitro model based on HUVEC cells, exposing them to H2O2 (400 µM) for 2 h.”

Could the author can be more accurate on the timing choice? Moreover, Did they consider to evaluate different timing?

Round 2

Reviewer 2 Report

The reviewer wish to thank the authors for accepting all of their suggestions and also to correct all the corrections proposed by the other reviewers.

Now the manuscript is in a convenient form to be published